# Automatic Gene Function Prediction in the 2020’s

**DOI:** 10.3390/genes11111264

**Published:** 2020-10-27

**Authors:** Stavros Makrodimitris, Roeland C. H. J. van Ham, Marcel J. T. Reinders

**Affiliations:** 1Delft Bioinformatics Lab, Delft University of Technology, 2628XE Delft, The Netherlands; r.c.h.j.vanham@tudelft.nl (R.C.H.J.v.H.); M.J.T.Reinders@tudelft.nl (M.J.T.R.); 2Keygene N.V., 6708PW Wageningen, The Netherlands; 3Leiden Computational Biology Center, Leiden University Medical Center, 2333ZC Leiden, The Netherlands

**Keywords:** automatic function prediction, Gene Ontology, protein representation, machine learning

## Abstract

The current rate at which new DNA and protein sequences are being generated is too fast to experimentally discover the functions of those sequences, emphasizing the need for accurate Automatic Function Prediction (AFP) methods. AFP has been an active and growing research field for decades and has made considerable progress in that time. However, it is certainly not solved. In this paper, we describe challenges that the AFP field still has to overcome in the future to increase its applicability. The challenges we consider are how to: (1) include condition-specific functional annotation, (2) predict functions for non-model species, (3) include new informative data sources, (4) deal with the biases of Gene Ontology (GO) annotations, and (5) maximally exploit the GO to obtain performance gains. We also provide recommendations for addressing those challenges, by adapting (1) the way we represent proteins and genes, (2) the way we represent gene functions, and (3) the algorithms that perform the prediction from gene to function. Together, we show that AFP is still a vibrant research area that can benefit from continuing advances in machine learning with which AFP in the 2020s can again take a large step forward reinforcing the power of computational biology.

## 1. Introduction

Automatic function prediction (AFP) deals with the algorithmic assignment of functional annotations—usually Gene Ontology (GO) terms—to proteins/genes of unknown function from proteins/genes whose function has already been determined experimentally. In the past two decades, the amount of new protein sequences has been growing at such a fast pace [1] that no experimental screen can keep up, making AFP a necessity for modern biology. In addition to generating fundamental biological knowledge about what proteins do, AFP is crucial for other aspects of research, such as linking genotype to phenotype, by enabling gene set enrichment analyses or facilitating the interpretation of GWAS hits (e.g., [2]). A far from exhaustive list of some of the recent successful AFP models is given in Table 1.

The Critical Assessment of Functional Annotation (CAFA) challenges provide an objective evaluation of modern AFP algorithms on a set of proteins with newly-acquired GO annotations [10,11,12]. The main finding of these challenges is that methods significantly improved between CAFA1 and CAFA2, but remained rather stagnant in CAFA3, with the exception of one novel method, GOLabeler [3], that outperformed all others, especially in the Molecular Function Ontology (MFO) [12]. Several methods performed rather similarly in the Biological Process Ontology (BPO) and all participating methods failed to outperform a simple co-expression-based baseline method at predicting cell motility and biofilm formation in *Pseudomonas aeruginosa* [12]. Together, these results show that the problem of AFP is far from solved and that perhaps one or several leaps are required to advance the field.

More than a decade ago, in a paper that set the foundation for the CAFA benchmarks, Godzik et al. defined three main challenges for AFP research [13]:How to extend AFP beyond homology transfer.How to define protein function in a standardized way.How to properly evaluate AFP methods.

Since then, all three of these questions have been addressed to varying extents. Several algorithms have been proposed which make use of different data sources, such as sequence features [14], gene expression, and protein-protein interactions [15]. The GO has become the standard vocabulary for describing protein function in the vast majority of AFP models, and the CAFA is a widely-accepted platform to objectively evaluate these models. To inspire the AFP field in realizing new breakthroughs, we have attempted to identify some (new) challenges that we feel are important for advancing the field and we will address them in detail in the next sections. These challenges are:How can we deal with biological function being tissue, cell-type, or condition-specific?How do we predict functions in non-model species?What data sources should be used for predicting function?How does missingness or bias in GO annotations affect the training of AFP models?How can we better exploit the Gene Ontology structure to improve functional annotation?

We will discuss these challenges from three different perspectives of an AFP pipeline (Figure 1): (a) how proteins are represented, (b) how function is encoded, and (c) what kind of prediction algorithms one should use. Protein representations refer to the input data types that feed an AFP model, e.g., sequence similarities or raw amino acid sequences as well as any feature extraction steps. Generally, functions are described using GO terms, but these have limitations or may require adaptations to cope with the challenges identified. Finally, prediction algorithms deal with the mapping between from input protein representation to the target function predictions.

## 2. Tissue/Condition-Specificity

Simply assigning a set of GO terms to a protein is often not enough. For instance, it does not necessarily provide information about whether the protein performs this function in specific tissues or under specific conditions, which is especially important for the Biological Process Ontology (BPO). For example, from gene expression experiments, it is known that genes can change co-expression partners [16] and regulators [17] from tissue to tissue [16], when under stress [18] or at different developmental stages [19]. Also, tissue-specific protein-protein interactions are known [20]. In other words, although a protein can be involved in multiple biological processes, it doesn’t have to execute all these functions at all times, which has two implications. On one hand, it will be difficult to validate predicted annotations when it is not known for which tissues or cell types the protein performs this function. On the other hand, we need to have tissue or cell-type specific information of the activity of the protein (such as mRNA expression levels) to be able to discover these functions. Greene et al. have demonstrated the importance of this issue, by constructing tissue-specific co-functional networks from existing GO annotations and tissue-specific gene expression information, leading to more accurate predictions of response to perturbation and discovery of gene-disease associations [21]. All of this is made even harder, as, at this point, there is not even a clear definition of a cell type. The Cell Ontology (CO) is a good step towards standardizing cell type definitions, but it is only restricted to animal cells [22]. Despite this, we believe that AFP researchers and curators of the GO need to start preparing for a possible transition to this more specialized AFP phase.

### 2.1. Protein Representation

To make protein representation tissue-specific, Zitnik and Leskovec adapted the node2vec method to extract tissue-specific protein embeddings from tissue-specific Protein-Protein Interaction (PP) networks which were then fed to a linear classifier to predict function [23]. It would be interesting to extend this approach to co-expression networks, which are probably a lot more variable. For cases where co-expression or interaction evolves over time, e.g., for developmental processes or stress responses, it might be helpful to look at dynamic network embedding algorithms, such as dynnode2vec [24] that can learn condition-specific node embeddings without prior knowledge and more efficiently than the approach of [23]. Advances in single-cell sequencing [25] and the generation of cell atlases [26,27] are expected to elucidate even more subtleties of protein function and thus provide a valuable resource for more fine-grained functional annotation. Being able to represent genes by their expression and/or methylation pattern [25] in millions of cells and not by the average of those quantities, as is done with bulk sequencing, can help us find rare but also specific gene functions. On the other hand, this creates computational challenges, as one single-cell experiment can nowadays generate data for millions of cells. Integrating data from multiple such experiments will require the use of techniques specialized for processing ‘big data’.

### 2.2. Function Representation

Unfortunately, predicting cell-type-specific and/or condition-specific function is a lot harder, as the number of possible outcomes increases in a combinatorial fashion. The number of GO terms is already very large, so creating a separate target variable for each combination of GO term, cell-type and condition would be intractable. Also, this would make the set of existing annotations even sparser posing severe problems for learning algorithms [28]. The Gene Ontology Consortium [29] uses so-called annotation extensions [30] to specify details about an annotation (including that the function occurs at a specific cell type) instead of creating a new GO term for each combination of function and cell type. However, the number of such extensions is also very large, so this does not solve the issue. Perhaps we are in need of a more fundamental representation of BPO functions. For example, for Molecular Function Ontology (MFO) terms, protein domains are often used as clear representatives of function, as specific domains correspond to specific 3D folding patterns that enable specific chemical reactions and therefore can be accurately associated with a molecular function. Certainly, domains can also be associated with BPO terms, but there rarely is a clear causal link. For instance, a DNA-binding domain might indicate that a protein could be a transcription factor, but that does not provide insights into the genes that this transcription factor regulates or the biological process(es) these genes are involved in. The introduction of Causal Activity Modelling (GO-CAM) [31] tries to address this issue and to unify GO terms by using causal graphs to model their interrelations. Alternatively, markers, such as DNA methylation, chromatin accessibility, and transcription factor binding can be used as a tissue-specific function representation, as they describe a gene’s regulation, which might provide information about its functions. The different data types can be integrated with appropriate approaches (e.g., [32]) to identify their common and independent components.

### 2.3. Prediction Methods

When the number of labels increases dramatically, model learning should also be adapted to handle this increase. A promising option would be to move from a discrete to a continuous representation of the conditions. Way and Greene have demonstrated this as a proof of principle by training a Variational Auto-Encoder (VAE) on gene expression data from different cancer types [33]. They then showed that the latent space learned during the unsupervised training contained directions that encoded important information such as gender, tissue of origin and presence or absence of a metastasis [33]. Further work needs to be done on the interpretation of such models, so that we can make use of the latent encoding of different conditions for predicting function. Since VAE’s are generative models, they could perhaps also generate predicted gene expression data for combinations of tissues and stresses that are not in the training set. Other generative models, such as Generative Adversarial Networks (GANs) [34] could also be used and specifically conditional GANs [35] are designed to generate data for a specific condition given a (discrete or continuous) numerical representation of that condition. We believe that this line of research will become popular in the near future. GANs have already been used in AFP to generate artificial data to counter class imbalance, thereby performing data augmentation [36] and leading to increased performance [37]. Incorporating the ‘grammar’ of GO-CAM relations into prediction models will also be an interesting challenge. For example, genes or proteins can also be viewed as part of the GO-CAM causal graph, thereby transforming the AFP problem into a semi-supervised learning problem of predicting new edges in that graph, specifically edges connecting the genes with the terms.

## 3. Going beyond Model Species

The need for accurate AFP is especially pressing for non-model species. Plants are interesting in that regard, as experimentally-derived functions in most plants are either very sparse or non-existent and the huge number of genes per species (e.g., more than 100,000 in wheat [38]) means that genome-wide experimental annotation would require vast amounts of time and resources. On the other hand, a lot of labeled data are required to train, as well as to test an AFP algorithm. Currently, labeled data come mostly from model species, as proteins from ten species account for 86–88% of the experimental GO annotations in UniProtKB, depending on the ontology (Figure 2).

One of the findings of the CAFA challenges [10,11,12] is that ensemble methods that combine predictions from many data sources tend to perform very well (e.g., MS-kNN [15] in CAFA2; and GOLabeler [3] and INGA [5] in CAFA3). Although CAFA is extremely useful, the evaluations rely on recent experimental annotations and these, by definition, are in their vast majority from 10–15 model species (Figure 3), because most experimental biologists work on those species. Besides plants, bacteria and archaea are also largely underrepresented in CAFA benchmarks (Figure 3). This focus on model species might hide the fact that perhaps some of the algorithms that are successful in CAFA might not be directly or fully applicable in non-model species, as for newly-sequenced species, typically, only DNA and protein sequences are available. We do by no means attempt to diminish the usefulness and impact of researching multi-omics ensemble methods, but it is important to realize that models that rely on protein-protein interactions or co-expression across different conditions are thus not applicable in the vast majority of species across all domains of life. This is especially important when predicting BPO terms, as these models have been shown to rely more on non-sequence-based data sources [12].

### 3.1. Protein Representation

A big question for AFP is whether sequence-only methods can achieve as high performance as methods that use multi-omic data. Recent work has shown that it is possible to accurately predict PPI’s from amino-acid sequence [39] or gene expression from DNA sequence [40], which implies that a big part of the multi-omic data is encoded in the sequence data, albeit in a more complicated manner. In other words, one could improve existing techniques that predict multi-omic data and use those predictions for AFP. But, this might also imply that sequence-only models might have the potential to perform equally well to the ensemble methods. Alternatively, as more and more species are being sequenced, finding orthologs or constructing high-quality multiple sequence alignments of proteins will get easier and easier. That, in turn, would imply more robust and accurate old-school, sequence-based annotation transfers with ever-increasing coverage. Essential wet-lab experiments could in that scenario be focused on the ever-decreasing set of proteins without close homologues.

### 3.2. Function Representation

Another big challenge of predicting protein functions in other species is that certain functions might be more prevalent or even unique in certain lineages. As an extreme example: it is not trivial to predict photosynthesis-related functions when training only on animal data. It would therefore be beneficial to have a function representation that allows extrapolating to new functions. To do so, we need our representation to be agnostic of the training annotations and capture more general functional aspects. This can be done by learning an embedding for each GO term so that terms that describe related functions have high similarity in the embedding space. That would enable us to extrapolate the meaning of terms beyond the training set. Several such representations have been proposed, one of the first ones being clusDCA [41], which used random walks to learn features that reflect the GO graph topology. More recent approaches make use of advances in Natural Language Processing (NLP) to learn embeddings that reflect semantic meanings based on the term names and/or descriptions [42]. Theoretical work has shown the utility of embedding graph-structured data (as GO terms are) in hyperbolic rather than Euclidean spaces [43].

### 3.3. Prediction Methods

Prediction models across species should mainly deal with two issues: (1) the presence of novel functions, as described above, and (2) the potential differences in distribution of the input data between species. There is a vast machine learning literature on few-shot and zero-shot learning, which deals with classification models that can make predictions for classes for which only very few or even no examples have been seen, respectively [44,45]. Such methods often tend to use class embeddings [46] and try to leverage prior knowledge on similarities between the classes. A similar approach has been applied in predicting novel cell types from gene expression data using Cell Ontology [22] embeddings [47]. Such approaches can additionally be useful for describing new terms that are occasionally added to the ontology, even before they accumulate many annotations. As for the difference in distributions, also known as domain shift [48], it can lead to large performance loss if not taken into account. As an example, methods that use the frequency of amino acids or amino acid n-grams in a sequence can be sensitive to amino acid frequency differences across lineages (e.g., [49]). Given certain assumptions about the type of domain shift, there exist different approaches for correcting it [48]. Even if theoretical assumptions are not met, however, domain adaptation can still give a performance boost (e.g., [50]), so even then it might still be worth attempting to detect and correct domain shifts.

## 4. Overlooked Data Sources

The amino acid sequence is the most widely-used data source for function prediction, followed by sequence-derived features such as domains as well as other omics data, such as gene expression or protein-protein interactions. There is, however, a wealth of other omics data that has the potential to predict function accurately, but that is currently hardly used. Leveraging these data could also boost performance of AFP algorithms. For the sake of brevity, we focus on three such omics data sources: proteomics, genome proximity, and epigenetic data. In addition, we also address the utility of literature mining.

### 4.1. Protein Representation

The power of gene expression data in function prediction has been well-documented in the CAFA challenges, especially for predicting BPO terms [12]. Interestingly, a study from 2017 found that co-expression networks constructed from proteomics rather than transcriptomics were more efficient at predicting both GO terms and pathways from the Kyoto Encyclopedia of Genes and Genomes (KEGG) in *Homo sapiens* [51]. Furthermore, mRNA and proteomics have been shown to contain complementary information [52,53,54] (Figure 4), caused by–amongst others–differences in degradation rates and translation speeds. Hence, protein abundances as measured, for example, by mass spectrometry give a more representative picture of the function of a protein than their mRNA proxies. Integrating these two omics data types could boost prediction accuracy. In addition, the relationship between genes and proteins is not 1-to-1, as RNA splicing [55] as well as post-translational modifications (PTM’s) [56] cause the encoding of multiple protein isoforms that have been shown to have different roles and functions [57]. This introduces an additional layer of complexity when trying to predict the function of genes, that can be partly addressed by measuring the expression of isoforms using RNAseq. However, different PTM’s can only be measured using proteomics techniques. As the quality, reproducibility, scalability and accessibility of mass spectrometry methods keep increasing [58], we expect proteomics to start playing a more and more prominent role in AFP.

Proximity between genes has also been shown to be indicative of co-functionality [59]. This is particularly true for bacteria, where genes from the same pathway are often organized in operons, but it was recently shown to be applicable in eukaryotes as well [60]. Except for linear proximity, the same holds for genes that are close in three-dimensional space, as these tend to be co-regulated [61] (Figure 4). Chromosome conformation data, generated using e.g., 4C [62] or Hi-C [63] techniques, have been used in a human protein function prediction pipeline [64], but they are certainly not exploited enough.

Epigenetic markers, such as DNA methylation or chromosome accessibility, affect gene regulation, implying that genes that have similar epigenetic patterns across tissues might be regulated jointly. This makes epigenetic data potentially a rich resource for predicting Biological Process terms, although it is not known to what extent this signal is complementary to gene expression. Human and mouse are the two species where this hypothesis can be easily tested as many of their genes have well-documented functions and the ENCODE project has generated large amounts of epigenetic data in both species [65].

Finally, mining of scientific literature has been under-used in the past years, but a recent study showed that it can have competitive performance [66]. The idea behind the usage of such text mining methods is that if two genes are often mentioned together in publications, then they are likely to be involved in the same function. In addition, co-occurrence of gene names with other words, such as disease or pathway names [67], can be informative. As successful NLP models are starting to be applied in biomedical literature data [67,68], we expect the role of text mining in AFP to increase in the immediate future.

### 4.2. Function Representation

Using new data sources in both computational and experimental protein annotation might call for new evidence codes. For instance, if indeed protein co-expression is much more relevant for co-functionality than mRNA co-expression, it might make sense to differentiate between functions discovered using the two technologies by splitting the HEP (High-throughput Expression Pattern) evidence code. For similar epigenetic profiles and co-regulation in general, there is also no appropriate evidence code. The closest one is perhaps IGI (Inferred from Genetic Interaction), but this mainly refers to one gene influencing another gene (e.g., by changes in expression or mutations), while co-regulation implies that both genes are jointly regulated by the same mechanism.

### 4.3. Prediction Methods

Arguably one of the most important recent methodological leaps is representation learning. Advances in machine learning have been quickly adopted by AFP researchers, causing a shift of the research efforts from the guilt-by-association (GBA) paradigm to automatic representation learning. A lot of recent works, often inspired by natural language processing, use convolutional or recurrent neural networks to automatically learn sequence features useful for predicting function. Such methods have been shown to work better than simple homology search (e.g., [69]). Also, several neural embedding methods have been recently proposed which can learn complex features for nodes of networks. Such methods have been applied to both PPI and co-expression networks and shown to be useful for reconstructing functional relationships [70,71]. However, there is still not enough evidence of whether such methods can outperform simple GBA methods, such as gene co-expression based on Pearson correlation, which is very effective for BPO predictions [12].

## 5. Biased and Missing Annotations

Another unresolved question that is important to address is how biases in GO annotations influence AFP. One extreme form of such a bias is missing annotations. Missing annotations in the test set do not have a dramatic effect on the ranking of AFP methods [72], but the effect of missing annotations in the training set has not yet been systematically quantified. We suspect that this effect is larger, especially for machine-learning-based methods that try to learn characteristics of proteins with the same function.

### 5.1. Protein Representation

What is often overlooked is that biases in the generation of annotations might also affect the protein representation. For example, in plant research, scientists are very often interested in stress responses and flowering, so a lot of the available gene expression data come from such conditions, meaning that it might be harder to infer other functions for which very little data are available. The same holds for other species. For example, *Drosophila melanogaster* is typically used to study genetics [73], *Caenorhabditis elegans* to study development [74] and so on. The detection of condition-specific or tissue-specific protein-protein interaction suffers from the same issue. Protein sequence data are also not completely ‘safe’ from this bias, as it is possible that a functional isoform of a protein used in rare, poorly-studied conditions is not known. However, for sequence data, the effect of that bias is arguably smaller than for other data types.

### 5.2. Function Representation

Many researchers tend to only include *experimental* evidence codes (EC’s) when training and testing AFP methods to avoid biases. This has the downside that a great amount of knowledge about protein function is potentially ignored. Moreover, experimental EC’s have also been shown to be highly biased [75], leading to the recent split between “experimental” and “high-throughput experimental” EC’s. Also, experimental EC’s include co-expression and protein interaction experiments which might introduce circular reasoning for algorithms that use such data types, in the same way that sequence similarity EC’s introduce bias for sequence-based algorithms. On the other hand, annotations that are automatically generated by a curated set of rules are labeled “IEA” and are often ignored, although they are rather reliable, given that the rules are made by expert curators [76]. We cannot convince experimentalists to start randomly selecting a protein and testing it for random functions to obtain an unbiased ground-truth with independent, identically distributed observations. Nor can we ever obtain a reliable, experimentally-derived set of negative annotations, as a protein might have a particular function only under certain conditions. Therefore, it might be interesting to try to further quantify the biases introduced by including certain EC’s. Previous work quantified the quality of automatic annotations by measuring to what extent they were later experimentally confirmed [76]. Additionally, one could look for possible changes in the performance of the naïve classifier when adding or removing annotations with a specific EC. Also, one could compare the performance of AFP methods for each evidence code separately to assess these biases better.

Some studies have attempted to tackle the missing annotations problem by generating negative examples [77,78]. Somewhat surprisingly, these datasets of negative annotations have not gained popularity among AFP researchers, as they are most often not used during the training of new methods. To our knowledge, there is no study providing evidence against using these data, so we believe that this topic deserves further investigation. A recent study showed that ignoring negative annotations at test time can lead to misleading evaluations [79].

### 5.3. Prediction Methods

Alternatively to generating negative annotations, we could also change the loss functions used to train AFP models. Most recent machine learning models are trained by minimizing a cross-entropy loss, which assumes no missing labels. That could be replaced by a loss for Positive-Unlabeled (PU) learning [80]. Again, these loss functions are well-known in AFP and related fields [77,81], but they seem not to be very popular. It is possible, that despite their obvious theoretical benefits, such losses do not work in practice for AFP and publication bias has prevented us from seeing these results. A different approach would be to introduce probabilistic labels, where annotations from EC’s that are considered reliable are attributed with high certainty, but unreliable ones (e.g., derived automatically by another AFP method and never verified by a curator) are used in the training set but with a low-probability. One could additionally always assign a non-zero probability to all other terms to account for missing annotations. Such a probabilistic ground-truth means that probabilistic learning algorithms will be needed, such as the graphical model proposed in [82].

## 6. Gene Ontology

GO is a very useful resource that describes biological function in a standardized manner that is human- and computer-readable. This has led to its nearly catholic acceptance as the go-to functional representation, to the point that function prediction is almost a synonym of GO term prediction.

### 6.1. Protein Representation

As stated above, automatic representation learning is a very promising direction. It can be further enhanced by unsupervised pre-training, where a generic protein representation is learned [83,84,85,86] from all available sequences. This representation can then be used to predict GO terms [87]. But end-to-end training is also possible, where the weights of the unsupervised feature extractor are also fine-tuned to create an ontology-specific feature representation designed for predicting GO terms from that ontology. This can lead to better performance, especially when only few labeled examples are available. Such approaches are currently the state-of-the-art in Natural Language Processing [88].

### 6.2. Function Representation

We previously touched upon embedding GO terms in a vector space and its potential in creating a new functional representation that is simpler but reflects the same semantic relationships as the GO graph. Figure 5 shows an example of such an embedding in two dimensions that reflects term co-occurence patterns. The biggest downside of those approaches is the loss of the interpretability and human-readability that GO terms have. It is still not trivial to always provide a biological interpretation for a given set of GO terms (even though visualization techniques are helping in that regard [89]), but it is still relatively easy to understand the meaning of an individual term by its name, description, and connections to ancestors and descendants. On the other hand, representing terms as high-dimensional vectors makes us lose the intuition, which implies that we would like this representation to be invertible, i.e., also provide us with a rule to convert a given vector in this “functional space” back to a term or a set of terms, ideally in a unique way. This is possible for linear mappings and we and others have worked on such approaches [90,91]. Linear approaches can capture simple relationships between terms such as co-occurrence or mutual exclusivity of a pair of terms [90], but might struggle to find more complicated relationships “hidden” either in the graph or in the semantics of terms. We therefore suspect that non-linear (e.g., neural) term embeddings are required to capture the whole structure. Nevertheless, we think that substantial emphasis and attention should be put on maintaining the interpretability of these models. This is nowadays also a hot topic in machine learning and computer vision [92,93,94], which has been dominated by neural networks in the past decade.

### 6.3. Prediction Methods

The three different ontologies contain correlated information. For example, “DNA binding” (MFO) co-occurs with “DNA-templated regulation of transcription” (BPO, ρ=0.54) and “nucleus” (CCO, ρ=0.35) (Figure 5). GO curators are well aware of these co-occurrences and have hand-crafted rules to automatically transfer such annotations. However, little computational work has been done to improve upon this rule-based system, although this seems like a very promising direction, especially for the Limited-Knowledge category of CAFA, where the goal is to predict the functions of proteins in one ontology using its functions in others. A possible approach would be to embed the terms of each ontology to a vector space and then try to align the three ontologies, with the aim of discovering new cross-ontology similarities that are not obvious to the curators and could be exploited by function prediction algorithms. In Onto2vec, the authors learned a joint embedding for terms from all three ontologies and proteins [95]. They used it to calculate protein similarities for the downstream task of protein interaction prediction, but it would also be very interesting to examine similarities in the embedding space between pairs of GO terms to identify potentially unknown correlations.

Over time, unannotated or partly annotated genes obtain new GO annotations, i.e., the ground-truth data that can be used to train AFP models changes. Therefore, it would be interesting to have models which can incorporate such new knowledge without having to re-train from scratch. This could be achieved by using online machine learning algorithms [96,97].

## 7. Evaluation of AFP Algorithms

Next to the challenges described above, the problem of properly evaluating AFP models, as already initially identified as one of the three main challenges by Godzik et al. [13], is still lingering. It has been shown that temporal hold-out evaluation strategies, like the CAFA challenges, give more realistic estimates of the performance on new unseen data than cross-validation [98]. Given the success of omics data in CAFA—and specifically in CAFA-π [12]—we believe that a GBA-based baseline should be included in future CAFA editions, for example a GBA based on PPI’s from e.g., the STRING database [99]. This is to be preferred over a co-expression-based baseline as for many species this data is not readily available.

By comparing the rankings of methods in CAFA3 [12] across five evaluation metrics, we found that most widely-used metrics are highly correlated, with the Semantic Distance [100] being slightly different from the rest (Figure 6). However, a more recent simulation study questioned the validity of these commonly used evaluation measures. The authors also proposed novel measures that more accurately reflect the quality of the predictions [101]. The complicated and biased nature of GO annotations can indeed lead to misleading evaluations, so having appropriate evaluation measures is essential to ensure that the field keeps going forward.

## 8. Conclusions

AFP remains one of the most challenging bioinformatics tasks and despite its growing interest and recent progress, it is far from being completely solved. Here, we addressed some of the current and future challenges of the field. In our view, significant breakthroughs are expected mainly from the use of neural embeddings (for describing both proteins and GO terms) and the use of new technologies, such as proteomics. The problem of predicting function for non-model species is possibly the most challenging, as it may still require generation of experimental data. However, as the community of AFP researchers is growing [12], we are optimistic that these challenges will soon be tackled.

## Figures and Tables

**Figure 1 genes-11-01264-f001:**
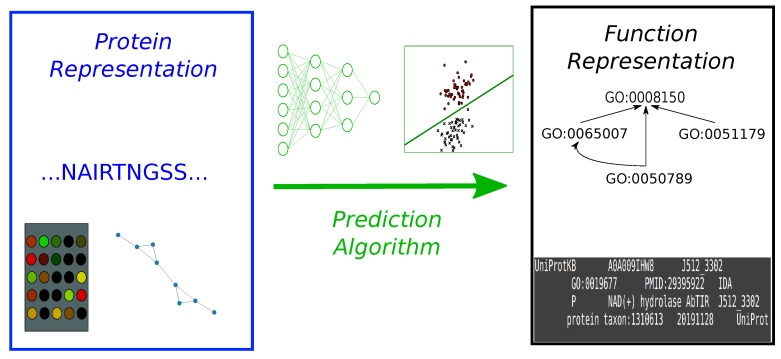
AFP algorithms typically consist of a protein representation (protein sequence, expression data, biological networks) (left, in blue), a function representation (often a vector of one-hot encoded GO terms) (right, in black) and a prediction algorithm that connects both (neural networks, support vector machines, Guilt-By-Association methods etc.) (middle, in green).

**Figure 2 genes-11-01264-f002:**
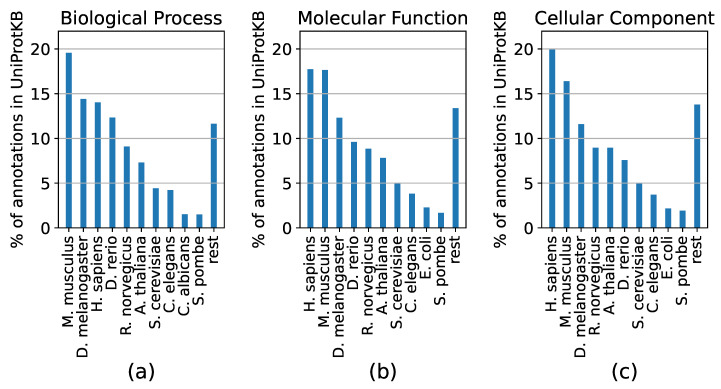
Distribution of experimental annotations from the Biological Process (**a**), Molecular Function (**b**) and Cellular Component (**c**) ontologies per species for proteins in UniProtKB. The ten species with the most annotations are shown for each ontology and annotations for all other species are shown in the ‘rest’ group.

**Figure 3 genes-11-01264-f003:**
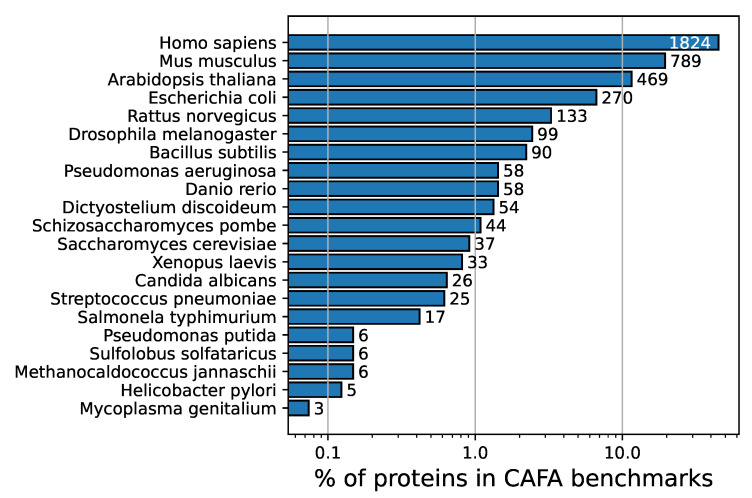
Percentage of proteins used in the three CAFA benchmarks [10,11,12] (*x*-axis, in log scale) per species. The absolute number of proteins per species is also given next to the bars. Only newly-annotated proteins are included, i.e., proteins that had no GO annotations before the benchmark (referred to as No-Knowledge benchmarks in CAFA). CAFA: Critical Assessment of Functional Annotation.

**Figure 4 genes-11-01264-f004:**
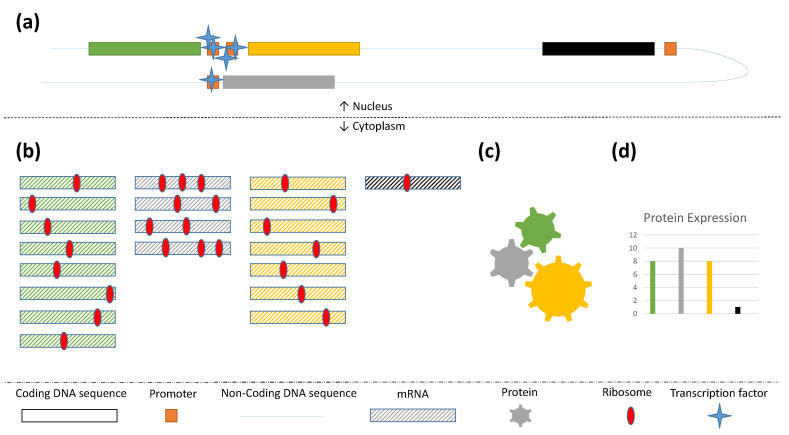
Genes that are close to each other in space are likely to be co-expressed, because transcription factors that bind on a promoter are likely to also bind to nearby promoters (**a**). Differences in expression are indicated by the amount of transcripts in the cytoplasm (**b**). In this case the green, yellow and gray genes code for proteins (shown as gears of the same color) that form a protein complex and perform a function together (**c**). The ribosome occupancy of the mRNAs is used to indicate differences in translation efficiency which result in similar abundances for proteins (**d**) despite different mRNA abundancies.

**Figure 5 genes-11-01264-f005:**
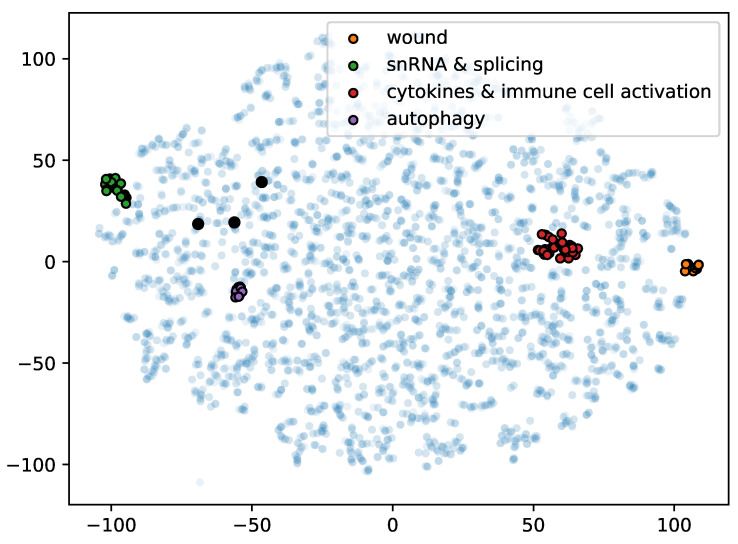
Two-dimensional tSNE embedding of GO terms from all three ontologies that annotate at least 0.1% of SwissProt entries. One minus the Pearson correlation of the occurence patterns of terms across SwissProt proteins was used as a distance measure for calculating the embeddings. Inspecting the terms in some of the clusters observed in this 2D space revealed that terms from the same cluster have similar meanings. Examples of clusters with terms that refer to wound healing, small nuclear RNAs and mRNA splicing complexes, cytokines and immune activation, and autophagy are shown in orange, green, red, and purple respectively. Terms “DNA binding”, “nucleus” and “DNA-templated regulation of transcription” are shown as larger black dots.

**Figure 6 genes-11-01264-f006:**
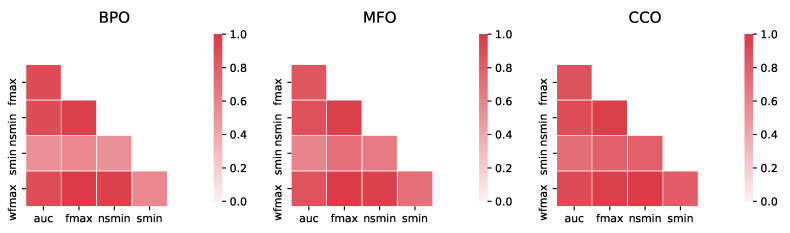
Pairwise absolute rank correlations between five different protein-centric evaluation metrics: area under the precision-recall curve (auc), maximum F1 score (fmax), maximum F1 score weighted by information content (wfmax), minimum semantic distance (smin), and minimum normalized semantic distance (nsmin). Correlation was calculated from the 146 methods that participated in CAFA3 [12] for the biological process (**left**), molecular function (**middle**) and cellular component (**right**) ontologies. More intense red color denotes larger absolute correlation. All pairwise correlations are statistically significant with uncorrected *p*-values $<10^{-11}$.

**Table 1 genes-11-01264-t001:** Some of the most important AFP models proposed in the past years.

Name	Reference	Input Data	Method
GOLabeler	[3]	Amino acid sequence, GO term frequencies	Learning to rank
FunFams	[4]	Amino acid sequence	Hidden Markov Model
INGA	[5]	Amino acid sequence	Homology search, enrichment analysis
PFP	[6]	Amino acid sequence	Phylogenetics
COFACTOR	[7]	Amino acid sequence, protein structure, protein interactions	Homology search, structural similarity
NetGO	[8]	Amino acid sequence, GO term frequencies, protein interactions	Learning to rank
DeepGOPlus	[9]	Amino acid sequence	Convolutional neural network, homology search

AFP: Automatic Function Prediction. GO: Gene Ontology.

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
