# Peer review of "Automatic Gene Function Prediction in the 2020’s"

_genes, 2020, doi:10.3390/genes11111264_

Round 1
Reviewer 1 Report
i) Authors should emphasize on the use of literature mining techniques to predict function of genes.
ii) It would be interesting to discuss how should researchers cope with the problem of frequent updates on the function of genes.
iii) Authors should discuss how mass spectrometry studies are adding a new role of strength to AFP.
iv) Does the use of guilt-by-association algorithm correlate with experimental follow ups? A detailed paragraph on the efficiency of this algorithms with respect to their conformation with experimental follow ups would be interesting.
v) A table showing the various softwares using these algorithms for AFP would be really interesting.
vi) How do the scoring schemes for AFP vary across algorithms using different approaches?
Author Response
First, we would like to thank the editor and reviewers for their constructive comments. They helped us to improve the manuscript considerably. We provide a point-by-point answer to each of them. We present answers to reviewer’s comments (in blue) in black and report changes to the manuscript in italic.
Apart from the changes suggested by the reviewers, we have also made two additional changes which we describe here:
1) In our discussion of GO term embeddings (lines 180-181), we mention that the GO graph structure could benefit from embedding terms in a hyperbolic space.
Changes in manuscript: “Theoretical work has shown the utility of embedding graph-structured data (as GO terms are) in hyperbolic rather than Euclidean spaces (Chamberlain et al. arxiv, 2017).”
2) We refer to work on GO Causal Activity Modelling (GO-CAM), which is an adapted function representation by the GO consortium, and mention challenges of dealing with this new representation.
Changes in manuscript: Section 1.2 (lines 109-114): “The introduction of Causal Activity Modelling (GO-CAM) (Thomas et al. Nature Genetics, 2019) tries to address this issue and to unify GO terms by using causal graphs to model their interrelations.”
Section 1.3 (lines 130-134): “Incorporating the 'grammar' of GO-CAM relations into prediction models will also be an interesting challenge. For example, genes or proteins can also be viewed as part of the GO-CAM causal graph, thereby transforming the AFP problem into a semi-supervised learning problem of predicting new edges in that graph, specifically edges connecting the genes with the terms.”
Comments to the authors
1) Authors should emphasize on the use of literature mining techniques to predict function of genes.
Literature mining is indeed an approach that has been widely overlooked in the past by the function prediction community, but a few recent works have shown that it can be very effective. We include it as an additional ‘overlooked data source’.
Changes in manuscript: We have added the following paragraph in section 3.1: “Finally, mining of scientific literature has been under-used in the past years, but a recent study showed that it can have competitive performance (You et al. Methods, 2018). The idea behind the usage of such text mining methods is that if two genes are often mentioned together in publications, then they are likely to be involved in the same function. In addition, co-occurrence of gene names with other words, such as disease or pathway names (Chen et al. PLOS Comput Biol, 2020), can be informative. As successful NLP models are starting to be applied in biomedical literature data (Lee at al. Bioinformatics 2019), we expect the role of text mining in AFP to increase in the immediate future.”
2) It would be interesting to discuss how should researchers cope with the problem of frequent updates on the function of genes.
The reviewer correctly points out that gene annotations are not static. The GO graph changes as terms are occasionally added, removed or renamed. The annotations also change as more knowledge is accumulated and new annotations are added to existing or new genes. For changes in the GO itself, we propose the use of GO term embeddings (either based on the graph topology or the textual description of terms) to identify similar terms, as this relates to our discussion of terms that are not present in a specific domain of life (lines 169-180). For the continuous accumulation of new annotations, one could think of an online learning scheme, where new data points are incorporated into an existing model without having to retrain from scratch.
Changes in manuscript: In section 2.3 (lines 190-191), we have added a sentence about the value of term embeddings with respect to newly-added terms: “Such approaches can additionally be useful for describing new terms that are occasionally added to the ontology, even before they accumulate many annotations.” At the end of section 5.3 (lines 367-370), we have added a paragraph about online learning: “Over time, unannotated or partly annotated genes obtain new GO annotations, i.e. the ground-truth data that can be used to train AFP models changes. Therefore, it would be interesting to have models which can incorporate such new knowledge without having to re-train from scratch. This could be achieved by using online machine learning algorithms.(Venkatesan et al. Evolving Systems, 2017;Ahmadi & Kramer arxiv, 2018)”
3) Authors should discuss how mass spectrometry studies are adding a new role of strength to AFP.
Protein abundances and RNA abundances do not correlate well for many genes (Vogel & Marcotte Nature Reviews Genetics, 2012). With the developments of mass spectometry, it becomes easier and more accurate to measure protein abundancs in high-throughput. The establishment of these protein abundances across tissues and conditions are thus providing a more realistic description of their function than their mRNA proxies.
Changes in manuscript: We have included the following sentence in section 3.1 (lines 211-216): “Furthermore, mRNA and proteomics have been shown to contain complementary information (Griffin et al. Molecular and Cellular Proteomics, 2002; Wang et al. Proteomics, 2014; Grabowski et al. Molecular and Cellular Proteomics, 2018) (Figure 4), amongst others caused by differences in degradation rates and translation speeds. Hence, protein abundances as measured, for example, by mass spectrometry give a more representative picture of the function of a protein than their mRNA proxies. Hence, integrating these two omics levels could potentially boost prediction accuracy.”
4) Does the use of guilt-by-association algorithm correlate with experimental follow ups? A detailed paragraph on the efficiency of this algorithms with respect to their conformation with experimental follow ups would be interesting.
The CAFA challenges evaluate the ability of methods to predict GO annotations that will be added in the future. In CAFA-π, where the goal was to predict genes involved in biofilm formation and motility in Pseudomonas aeruginosa, a guilt-by-association (GBA) baseline based on gene co-expression achieved the best performance among all participating methods. This shows that indeed GBA algorithms can indeed correlate well with future experiments, which might suggest that it is a good idea to have such a baseline in future CAFA challenges. Even more so, as a baseline based on co-expression (which is now often used) is impractical because for many species this data is not available.
Changes in manuscript: We have added the suggestion of a GBA-based baseline in the ‘evaluation’ section (lines 375-379): “Given the success of omics data in CAFA - and specifically in CAFA-π - we believe that a GBA-based baseline should be included in future CAFA editions, for example a GBA based on PPI's from e.g. the STRING database. This is to be preferred over a co-expression-based baseline as for many species this data is not readily available.”
5) A table showing the various softwares using these algorithms for AFP would be really interesting.
Changes in manuscript: In our introduction section, we have included a Table (Table 1 shown below) with details about some models that did well in CAFA3, including references and a short description. We have also included some promising new methods that are too new to have participated in the CAFA3 challenge.
6) How do the scoring schemes for AFP vary across algorithms using different approaches?
We collected the performances of 146 methods that participated in CAFA3 that were evaluated using the protein centric Area Under the Precision Recall Curve (), maximum F1 score (), maximum F1 score weighted by term information content (), minimum Semantic Distance () and minimum normalized Semantic distance (). To see whether these five scoring schemes give similar rankings of the methods, we calculated the Spearman correlation between all pairs of metrics over the 146 observations. Note that for , and
, higher values mean better performance, but for and it is the other way around, meaning that a correlation of -1 between e.g. and would imply identical ranking. We therefore used the absolute value of the correlation. We repeated that for the three ontologies and the results are summarized in the figure below. As we can see, four of the metrics give very similar rankings and only is considerably different.
Changes in manuscript: We have added this figure (as Figure 6) in the evaluation section. We have also modified the text in that section (lines 380-383) as follows: “By comparing the rankings of methods in CAFA3 across five evaluation metrics, we found that most widely-used metrics are highly correlated, with the Semantic Distance being slightly different from the rest (Figure 6). However, a more recent simulation study questioned the validity of these commonly used evaluation measures. The authors also proposed novel measures that more accurately reflect the quality of the predictions.”

Reviewer 2 Report
Dear authors,
The presented MS "Automatic Gene Function Prediction in the 2020’s" is a detailed review of the current status of the field and addresses future directions for an automated learning of gene function prediction. It nicely summarises recent developments/trends and addresses options for improvement in a clearly structured way. It is a really informative manuscript and I enjoyed reading it.
I believe it is of profound general interest to the scientific community. I am listing some suggestions that came to my mind while reading the manuscript. The comments are all minor and I leave it to the discretion of the authors whether to include them or not. Finally, I added in a short list of suggested text changes and hope mot make the manuscript ever so slightly better.
COMMENTS and SUGGESTIONS:
Figure Legend to Figure 1 is slightly confusing? I would change it to something like that: Figure 1. AFP algorithms typically consist of a protein representation (protein sequence, expression data, biological networks) (left, in blue), a function representation (often a vector of one-hot encoded GO terms) (right, in black) and a prediction algorithm that connects both (neural networks, support vector machines, Guilt-By-Association methods etc.) (middle, in green).
Lines 84-86: The authors refer to single cell analyses and cell atlas programmes. Those are major initiatives that provide a huge amount of additional data in respect to gene function in a cell specific and highly spatiotemporal approach. I would make a stronger point of those datasets that are so far not really integrated, presumably due to the sheer amount of data that needs to be considered. Would it be worthwhile to extend this line of thought a bit further and potentially discuss whether it can be implemented or discuss future initiatives for protein representation (as discussed in part 1.1)? I know this is considered further down (e.g. lines 147 and 148), however, I am sure a repeated consideration would not be bad and will illustrate the complexity even further.
Line 105: Would it be worthwhile mentioning integration of ATAC, 3C/4C or ChIP-Seq data integrating DNA binding transcription factors for function representation? I am aware that also this set of data is immense, however, a speculation on how to integrate that might be nice at this point?
Lines 120/121: You mention how much GANs have helped avoiding the imbalance. Maybe you could give some experimental examples to showcase what it did and why it was crucial to apply it? Just to get a better grip on the importance thereof?
Line 195: Is it worthwhile mentioning splice isoforms to make the point even more clear how important in the end proteomics is? I think it cannot be overstressed… ;-)
Lines 204 following: Mentioning of conformation capture approaches and trying to integrate those data as well? Just a thought, it would fit nice into this section (see also comment to line 105).
Figure 5: Is there a way to shade the tSNE entries slightly brighter (lighter even blue) to allow better visualization of the relevant dots?
CHANGES TO BE MADE:
Lines 35-37: Potentially use bullet points or letters instead of numbers. Two subsequent numberings (lines 35-37 and 45-49) is slightly confusing.
Line 44: Potentially insert a link to the rest of the manuscript “and we will address those points in detail in the next 5 sections of our review.”
Line 53: omit full stop
Line 73: PPI write out what it stands for (i.e. Protein-Protein Interaction)
Line 103: leave out first comma
Line 125: one space too much between “e.g.” and “more”
Line 142: BP terms. This should read BPO terms?
Line 168: omit “that predict” to make it read: Prediction models across species…
Line 193: Describe the abbreviation “KEGG” (i.e. Kyoto Encyclopaedia of Genes and Genomes)
Line 274: consider removing the word “very” (unless VERY much justified by the quoted reference 62)
Line 344: Omit “To conclude,”, this is already introduced by the header.
Author Response
First, we would like to thank the editor and reviewers for their constructive comments. They helped us to improve the manuscript considerably. We provide a point-by-point answer to each of them. We present answers to reviewer’s comments (in blue) in black and report changes to the manuscript in italic.
Apart from the changes suggested by the reviewers, we have also made two additional changes which we describe here:
1) In our discussion of GO term embeddings (lines 180-181), we mention that the GO graph structure could benefit from embedding terms in a hyperbolic space.
Changes in manuscript: “Theoretical work has shown the utility of embedding graph-structured data (as GO terms are) in hyperbolic rather than Euclidean spaces (Chamberlain et al. arxiv, 2017).”
2) We refer to work on GO Causal Activity Modelling (GO-CAM), which is an adapted function representation by the GO consortium, and mention challenges of dealing with this new representation.
Changes in manuscript: Section 1.2 (lines 109-114): “The introduction of Causal Activity Modelling (GO-CAM) (Thomas et al. Nature Genetics, 2019) tries to address this issue and to unify GO terms by using causal graphs to model their interrelations.”
Section 1.3 (lines 130-134): “Incorporating the 'grammar' of GO-CAM relations into prediction models will also be an interesting challenge. For example, genes or proteins can also be viewed as part of the GO-CAM causal graph, thereby transforming the AFP problem into a semi-supervised learning problem of predicting new edges in that graph, specifically edges connecting the genes with the terms.”
Reviewer 2
The presented MS "Automatic Gene Function Prediction in the 2020’s" is a detailed review of the current status of the field and addresses future directions for an automated learning of gene function prediction. It nicely summarises recent developments/trends and addresses options for improvement in a clearly structured way. It is a really informative manuscript and I enjoyed reading it. I believe it is of profound general interest to the scientific community. I am listing some suggestions that came to my mind while reading the manuscript. The comments are all minor and I leave it to the discretion of the authors whether to include them or not. Finally, I added in a short list of suggested text changes and hope mot make the manuscript ever so slightly better.
Minor comments
1) Figure Legend to Figure 1 is slightly confusing? I would change it to something like that: Figure 1. AFP algorithms typically consist of a protein representation (protein sequence, expression data, biological networks) (left, in blue), a function representation (often a vector of one-hot encoded GO terms) (right, in black) and a prediction algorithm that connects both (neural networks, support vector machines, Guilt-By-Association methods etc.) (middle, in green).
Changes in manuscript: We have updated the legend accordingly.
2) Lines 84-86: The authors refer to single cell analyses and cell atlas programmes. Those are major initiatives that provide a huge amount of additional data in respect to gene function in a cell specific and highly spatiotemporal approach. I would make a stronger point of those datasets that are so far not really integrated, presumably due to the sheer amount of data that needs to be considered. Would it be worthwhile to extend this line of thought a bit further and potentially discuss whether it can be implemented or discuss future initiatives for protein representation (as discussed in part 1.1)? I know this is considered further down (e.g. lines 147 and 148), however, I am sure a repeated consideration would not be bad and will illustrate the complexity even further.
We agree with the reviewer that the use of single-cell data in AFP will create new challenges as we can now measure expression over several million cells per experiment, leading to an extensive amount of data to be handled.
Changes in manuscript: We have added the following sentence at the end of section 1.1 (lines 89-92): “On the other hand, this creates computational challenges, as one single-cell experiment can nowadays generate data for millions of cells. Integrating data from multiple such experiments will require the use of techniques specialized for processing 'big data'.”
3) Line 105: Would it be worthwhile mentioning integration of ATAC, 3C/4C or ChIP-Seq data integrating DNA binding transcription factors for function representation? I am aware that also this set of data is immense, however, a speculation on how to integrate that might be nice at this point?
That is a very interesting suggestion. Epigenetic markers, chromatin accessibility and the three-dimensional conformation of the genome say a lot about the regulatory pattern of genes and therefore their function. This means that they could be used as a function representation instead of/alongside GO terms. The different data types could be integrated by approaches such as JIVE (joint and individual variation explained) to discover the commonalities and differences between them.
Changes in manuscript: We have added the following (lines 110-114): “Alternatively, markers, such as DNA methylation, chromatin accessibility, and transcription factor binding can be used as a tissue-specific function representation, as they describe a gene's regulation, which might provide information about its functions. The different data types can be integrated with appropriate approaches (e.g.(Lock et al. Ann. App. Stat., 2013)) to identify their common and independent components.”
4) Lines 120/121: You mention how much GANs have helped avoiding the imbalance. Maybe you could give some experimental examples to showcase what it did and why it was crucial to apply it? Just to get a better grip on the importance thereof?
The improvement by GANs find its origin in their ability to generate artificial samples, which as such can be used as data augmentation, thereby having the potential to counter the class imbalance as well as having a small number of training samples.
Changes in manuscript: We adapted the text to further explain the effect of GANs in that work (lines 129-130): “GANs have already been used in AFP to generate artificial data to counter class imbalance, thereby performing data augmentation and leading to increased performance”
5) Line 195: Is it worthwhile mentioning splice isoforms to make the point even more clear how important in the end proteomics is? I think it cannot be overstressed… ;-)
It is indeed true that isoforms add an additional layer of complexity to gene function prediction as one gene sequence can code for multiple isoforms, i.e. multiple different proteins that do indeed have can have different functions. Most of these isoforms can be measured using RNAseq although post translational modifications can also introduce novel isoforms, indeed stressing the importance of proteomics.
Changes in manuscipt: We added the following sentence in lines 216-221: “Although genes code for proteins, their relationship is not 1-to-1, as RNA splicing (Wang et al. Methods, 2020) as well as post-translational modifications (PTM’s) (Perchey et al. Scientific Reports, 2019) cause the encoding of multiple protein isoforms that have been shown to have different roles and functions (Cszimok et al. Current Opinion in Structural Biology, 2018). This introduces an additional layer of complexity when trying to predict the function of genes, that can be partly addressed by measuring the expression of isoforms using RNAseq, whereas different PTM’s can be measured using proteomics techniques.”
6) Lines 204 following: Mentioning of conformation capture approaches and trying to integrate those data as well? Just a thought, it would fit nice into this section (see also comment to line 105).
Changes in manuscript: We now mention 4C and Hi-C as the two most common methods to assess genomic proximity in 3D space (line 228).
7) Figure 5: Is there a way to shade the tSNE entries slightly brighter (lighter even blue) to allow better visualization of the relevant dots?
Changes in manuscript: We have updated the figure as per the reviewer’s suggestion. Due to use of a different random seed while re-running the tSNE, the figure looks slightly different, but similar patterns appear.
Suggested text changes
8) Lines 35-37: Potentially use bullet points or letters instead of numbers. Two subsequent numberings (lines 35-37 and 45-49) is slightly confusing.
9) Line 44: Potentially insert a link to the rest of the manuscript “and we will address those points in detail in the next 5 sections of our review.”
10) Line 53: omit full stop
11) Line 73: PPI à write out what it stands for (i.e. Protein-Protein Interaction)
12) Line 103: leave out first comma
13) Line 125: one space too much between “e.g.” and “more”
14) Line 142: BP terms. This should read BPO terms?
15) Line 168: omit “that predict” to make it read: Prediction models across species…
16) Line 193: Describe the abbreviation “KEGG” (i.e. Kyoto Encyclopaedia of Genes and Genomes)
17) Line 274: consider removing the word “very” (unless VERY much justified by the quoted reference 62)
18) Line 344: Omit “To conclude,”, this is already introduced by the header.
Thank you for these suggestions.
Changes in manuscript: We have implemented these changes.

Reviewer 3 Report
The review by Makrodimitris and colleagues covers the current state-of-the-art of Automatic Gene Function prediction methods. The work is very well written, covers the subject matter both broadly and comprehensively and brings to attention important recent development in the field, all the while noting the challenges that are still outstanding. I can only say that I think the review is very good and it was a real pleasure to read.
Author Response
First, we would like to thank the editor and reviewers for their constructive comments. They helped us to improve the manuscript considerably. We provide a point-by-point answer to each of them. We present answers to reviewer’s comments (in blue) in black and report changes to the manuscript in italic.
Apart from the changes suggested by the reviewers, we have also made two additional changes which we describe here:
1) In our discussion of GO term embeddings (lines 180-181), we mention that the GO graph structure could benefit from embedding terms in a hyperbolic space.
Changes in manuscript: “Theoretical work has shown the utility of embedding graph-structured data (as GO terms are) in hyperbolic rather than Euclidean spaces (Chamberlain et al. arxiv, 2017).”
2) We refer to work on GO Causal Activity Modelling (GO-CAM), which is an adapted function representation by the GO consortium, and mention challenges of dealing with this new representation.
Changes in manuscript: Section 1.2 (lines 109-114): “The introduction of Causal Activity Modelling (GO-CAM) (Thomas et al. Nature Genetics, 2019) tries to address this issue and to unify GO terms by using causal graphs to model their interrelations.”
Section 1.3 (lines 130-134): “Incorporating the 'grammar' of GO-CAM relations into prediction models will also be an interesting challenge. For example, genes or proteins can also be viewed as part of the GO-CAM causal graph, thereby transforming the AFP problem into a semi-supervised learning problem of predicting new edges in that graph, specifically edges connecting the genes with the terms.”
Reviewer 3
The review by Makrodimitris and colleagues covers the current state-of-the-art of Automatic Gene Function prediction methods. The work is very well written, covers the subject matter both broadly and comprehensively and brings to attention important recent development in the field, all the while noting the challenges that are still outstanding. I can only say that I think the review is very good and it was a real pleasure to read.
Thank you for your nice comments.

Round 2
Reviewer 1 Report
The authors have addressed all the concerns with satisfactory explanation.